

# Monitoring cyanobacterial toxins in a large reservoir: relationships with water quality parameters

Seenivasan Subbiah,  Adcharee Karnjanapiboonwong,  Jonathan D. Maul, Degeng Wang and  Todd A. Anderson

Department of Environmental Toxicology, Texas Tech University, Lubbock, TX, United States of America

## ABSTRACT

Cyanobacteria are widely distributed in fresh, brackish, and ocean water environments, as well as in soil and on moist surfaces. Changes in the population of cyanobacteria can be an important indicator of alterations in water quality. Metabolites produced by blooms of cyanobacteria can be harmful, so cell counts are frequently monitored to assess the potential risk from cyanobacterial toxins. A frequent uncertainty in these types of assessments is the lack of strong relationships between cell count numbers and algal toxin concentrations. In an effort to use ion concentrations and other water quality parameters to determine the existence of any relationships with cyanobacterial toxin concentrations, we monitored four cyanobacterial toxins and inorganic ions in monthly water samples from a large reservoir over a 2-year period. Toxin concentrations during the study period never exceeded safety limits. In addition, toxin concentrations at levels above the limit of quantitation were infrequent during the 2-year sampling period; non-detects were common. Microcystin-LA was the least frequently detected analyte (86 of 89 samples were ND), followed by the other microcystins (microcystin-RR, microcystin-LR). Cylindrospermopsin and saxitoxin were the most frequently detected analytes. Microcystin and anatoxin concentrations were inversely correlated with $Cl^-$, $SO_4^{-2}$, $Na^+$, and $NH_4^+$, and directly correlated with turbidity and total P. Cylindrospermopsin and saxitoxin concentrations in water samples were inversely correlated with $Mg^{+2}$ and directly correlated with water temperature. Results of our study are expected to increase the understanding of potential relationships between human activities and water quality.

## INTRODUCTION

Anthropogenic pollution of freshwaters has altered microbiota, leading to changes in their functions, deterioration of water quality, and economic loss (*Carpenter et al., 1998*). Algae are a vital group in aquatic ecosystems and a significant factor for monitoring water quality. Algae are important indicators of ecosystem conditions because of their quick response to physicochemical changes in freshwater systems. Loading of inorganic nitrogen (N) and phosphorous (P) nutrients in aquatic systems contributes to increasing rates of cyanobacterial hazardous algal blooms (cHABs). Negative effects of cyanobacterial toxins

Corresponding author
Todd A. Anderson,
todd.anderson@ttu.edu

in rats, mice, and fish, as well as their detection in blue–green algae food supplements have been reported (*Kondo et al., 1996*; *Bruno et al., 2006*; *Drobac et al., 2016*). The economic loss due to cHABs in the United States are annually estimated at more than one billion dollars (*Dodds et al., 2009*). Human-induced eutrophication and subsequent algal blooms negatively impact recreational water use, waterfront property values, threatened and endangered species, and drinking water.

Data from the Harmful Algal Bloom-Related Illness Surveillance System (HABISS) from 2007–2011, which were summarized in *Backer et al. (2015)* indicated that, among 2,323 samples, cyanobacteria (73%) were the most common type of organism reported followed by *Anabaena spp.* (20% of samples), *Aphanizomenon spp.* (7% of samples), and *Microcystis spp.* (7% of samples). There are some reports available on factors affecting cyanobacterial growth (*Robarts & Zohary, 1987*). These include: effects of temperature on algal growth (*Butterwick, Heaney & Talling, 2005*), nitrate uptake capabilities of cyanobacteria (*Flores et al., 2005*), effects of temperature on algal distribution (*Breitbarth, Oschlies & LaRoche, 2007*), temporal variations in the photosynthetic activity of cyanobacteria species (*Hodoki et al., 2011*), and the influence of N, P, and other environmental factors on algal blooms in freshwater and marine systems (*Paerl, 2008*). *Gobler et al. (2016)* reported reductions of N and P were essential to reducing the intensity and toxicity of algal blooms in Lake Erie, however, *Harke et al. (2016)* suggested that blooms in western Lake Erie were not significantly reduced as a result of P nutrient management plans. Reports are also available on cyanobacterial toxins in lakes, reservoirs, and recreational sites in other countries (*Tsuji et al., 1996*; *Lehman et al., 2005*; *Dos Anjos et al., 2006*; *Bullerjahn et al., 2016*; *Francy et al., 2016*; *Loftin et al., 2016*; *Douma et al., 2017*).

In an effort to understand the potential impact of man-made activities and environmental conditions in the study location on water quality, we attempted to correlate water quality parameters measured in our own laboratory or obtained from the USEPA's STOrage and RETrieval (*EPA, 2016*) and Water Quality eXchange databases with cyanobacterial toxin concentrations in a reservoir.

# MATERIALS AND METHODS

## Chemicals and consumables

Methanol, acetonitrile, formic acid, ammonium formate, and water (all LC/MS or Optima grade) were obtained from Fisher Chemicals (Fair Lawn, NJ, USA). Ammonium hydroxide (NH$_3$ content 28–30%; analytical grade) was from Sigma-Aldrich (St. Louis, MO). Solid phase extraction (SPE) cartridges were 60-mg Oasis HLB with 30-$\mu$m particle size (Waters, Milford, MA). Borosilicate glass tubes, autosample vials, and polypropylene centrifuge tubes were purchased from Fisher Scientific. Milli Q water (>18 M$\Omega$) was produced from a water purification system (Barnstead Nanopure). Multi-anion standards were from Sigma-Aldrich (St. Louis, MO, USA) and multi-cation standards were from Alltech (USA). Anatoxin-a, microcystin-LA, -LR, and -RR standards, as well as saxitoxin and cylindrospermopsin ELISA kits were all from Abraxis (Warminster, PA, USA). The 25-mm glass microfiber GF/A syringe filters were obtained from Whatman (GE Healthcare UK

Limited, UK) and B-D 10-mL disposable syringes were from Becton Dickinson & Co (Franklin Lakes, NJ, USA). The IC eluents were sodium hydroxide (50%, w/w) for anion analysis and methanesulfonic acid for cation analysis.

## Sample collection

Water samples were collected from nine different locations within a recreational reservoir in the southwest US at monthly intervals from September 2015 to September 2017. The lake is 89,000 acres in area and holds >2.5 million acre-ft of water. The minimum distance between sample locations was one mile. The collected 500-mL water samples were stored in airtight amber containers and kept on dry ice until transport to our laboratories in Lubbock, TX within 24 h.

## Sample pre-treatment

The analysis of algal toxins in collected water was initiated immediately after receipt of samples. Frozen water samples were thawed, then sonicated for 3 h. An aliquot of each sample was collected for use while the remaining sample volume was stored in the freezer at −20 °C prior to inorganic ion analysis.

## Saxitoxin and cylindrospermopsin analysis by ELISA

For the ELISA tests, we followed the methods provided by the manufacturer (*Abraxis Inc, 2019a*; *Abraxis Inc, 2019b*). The 96-well plates were tested using a UV–VIS spectrophotometer (Spectra Max Plus; Molecular Devices, San Jose, CA, USA) at 450 nm.

## Anatoxin-a, Microcystin-LR, -LA, and -RR determination by LC-MS/MS

20 mL of pre-filtered water sample was aliquoted into a 50-mL polypropylene tube. In parallel, a laboratory blank sample (Milli-Q water), a laboratory control sample (LCS; spiked Milli-Q water), and a matrix-spike sample (MS) were also prepared for quality control. LCS and MS were spiked with Anatoxin-a, Microcystin -LR, -LA, and -RR each at 0.5 ng/mL. 0.5 mL of methanol and 0.1 mL of ammonium hydroxide were added to samples. Samples were vortexed and centrifuged at 3,000 rpm for 15 min. Oasis HLB cartridges (Waters) were conditioned using 5 mL of methanol followed by 5 mL of water. Samples were passed through the cartridges at a flow rate of 2–3 mL/min. HLB cartridges were then dried under high vacuum for 20 min. Cartridges were eluted 3X with 0.7 mL of 2% formic acid in methanol. Eluates were collected in glass tubes, evaporated under nitrogen, and reconstituted to 0.2 mL with 0.19 mL of methanol and 0.01 mL of 10 µg/mL simeton as an internal standard. The method used in this study followed *Cong et al. (2006)* and *Waters Corporation (2014)*. Spike-recovery tests with algal toxins met QA criteria (50%–150%) as described by *Shoemaker, Tettenhorst & De la Cruz (2015)*.

Algal toxins in samples were analyzed using a Thermo Scientific Accela ultra–high–performance liquid chromatography (UHPLC) system equipped with a Thermo Scientific TSQ Quantum Access Max triple stage quadrupole mass spectrometer (MS/MS) operated in heated electrospray ionization (HESI probe) mode with positive polarity. One µg/mL algal toxin standards were used for optimization to achieve consistent counts per second (cps) in MS (Q1) and MS/MS (Q3) scans. The parent and product ion transitions and other mass

spectrometry operating parameters are in Tables S1 and S2. Chromatographic separation was carried out using an Accela 1,250 binary pump, a CTC PAL autosampler system, a Phenomenex Nx 3 μ 150 mm × 2 mm diameter analytical column with a SecurityGuard ULTRA guard column. The column oven temperature was set to 30 °C. Elution solvents were 5 mM ammonium formate and 0.1% formic acid in water:acetonitrile (95:5) (A) and 5 mM ammonium formate and 0.1% formic acid in acetonitrile:methanol (50:50) (B). The mobile phase composition (A:B; v/v) was 90:10 at 0 min, 90:10 at 3 min, 10:90 at 6 min, 10:90 at 10 min, and 90:10 from 12 to 15 min with a flow rate of 300 μL/min and an injection volume of 20 μL. The LC-MS/MS instrument software was Xcalibur version 2.1.0.

### Anions and cations by ion chromatography (IC)

Water samples were diluted 1:5 with Milli-Q water, filtered (0.45 μ) and transferred to IC poly vials with filter caps. Simultaneously, a laboratory blank sample (Milli-Q water) and an LCS (Milli-Q water spiked with anions and cations) were prepared for quality control. Samples were analyzed by Dionex IC 25 Ion Chromatograph (Chromeleon software version CM 6.5 SP1) coupled with a conductivity detector (*Hautman, Munch & Pfaff, 1997*). Target ions were fluoride ($F^-$), chloride ($Cl^-$), bromide ($Br^-$), nitrate ($NO_3^-$), sulfate ($SO_4^{2-}$), sodium ($Na^+$), ammonium ($NH_4^+$), potassium ($K^+$), magnesium ($Mg^{2+}$), and calcium ($Ca^{2+}$). The column and instrumental parameters are presented in Supplemental Information (Table S3). Spike-recovery tests with test analytes met QA criteria (85%–115%) as described in U.S. EPA Method 300.1 (*Hautman, Munch & Pfaff, 1997*).

### Data analysis

Inorganic ion data for water samples were normally distributed, however, data for algal toxins were not normal and followed a typical pattern (logarithmic) for environmental residues where non-detects are frequent. The *p*-values for correlation analyses were determined by either ANOVA or the Mann–Whitney–Wilcoxon test in R.

## RESULTS AND DISCUSSION

### Cyanobacterial toxins in surface water

A summary of the results obtained from the analysis of freshwater lake samples for cyanobacterial toxins is presented in Table 1. Toxin concentrations during the study period never exceeded safety limits (0.3–1.6 μg/L for microcystins and 0.7–3 μg/L for cylindrospermopsin). Toxin concentrations at levels above the limit of quantitation were infrequent during the 2-year sampling period; non-detects (ND) were common. Microcystin-LA was the least frequently detected analyte (86 of 89 samples were ND), followed by the other microcystins (microcystin-RR: 80 of 89 samples were ND, microcystin-LR: 77 of 89 samples were ND). Cylindrospermopsin and saxitoxin were the most frequently detected analytes above the detection limit. Although saxitoxin was frequently detected, it was never above the reporting limit or quantitation limit (0.025 μg/L) in any sample.

**Table 1   Summary data for cyanobacterial toxin concentrations (μg/L) in 89 water samples collected from a reservoir in the southwest U.S. from September 2015 to September 2017.**

| Analyte | Mean | SD | Max | RL | # non-detects | # >RL | # <RL |
|---|---|---|---|---|---|---|---|
| Anatoxin-A[a] | 0.013 | 0.021 | 0.14 | 0.02 | 53 | 9 | 27 |
| M-RR[b] | 0.002 | 0.010 | 0.07 | 0.05 | 80 | 3 | 6 |
| M-LR[b] | 0.004 | 0.011 | 0.06 | 0.05 | 77 | 5 | 7 |
| M-LA[b] | 0.003 | 0.025 | 0.24 | 0.05 | 86 | 1 | 2 |
| Cylindrospermopsin[c] | 0.044 | 0.032 | 0.10 | 0.10 | 24 | 15 | 50 |
| Saxitoxin[d] | 0.017 | 0.012 | NA | 0.05 | 30 | 0 | 59 |

**Notes.**

SD,  standard deviation;  RL,  reporting limit.

[a]In calculating the mean, we used the detection limit (0.025 μg/L) for samples containing anatoxin at <RL and 0 for non-detect samples.

[b]In calculating the mean, we used the detection limit (0.01 μg/L) for samples containing microcystins at <RL and 0 for non-detect samples.

[c]In calculating the mean, we used $\frac{1}{2}$ the RL (0.05 μg/L) for samples containing cylindrospermopsin at <RL and 0 for non-detect samples.

[d]In calculating the mean, we used $\frac{1}{2}$ the RL (0.025 μg/L) for samples containing saxitoxin at <RL and 0 for non-detect samples.

## Relationships between water quality parameters and cyanobacterial toxins

Results of correlation analyses between water parameters and cyanobacterial toxin concentrations are presented in Table 2. Anatoxin-A concentrations in water samples were inversely related to $F^-$, $Cl^-$, $Br^-$, $NO_3^-$, $SO_4^{-2}$, $Na^+$, $NH_4^+$, conductivity, total dissolved solids (TDS), and dissolved oxygen (DO), but directly related to $Ca^{+2}$, pH, water temperature, turbidity, and total phosphorus. Microcystin-LR concentrations were the least correlated with water parameters; M-LR was inversely related to alkalinity and directly related to total suspended solids (TSS), turbidity, and total phosphorus. Microcystin-LA concentrations in water samples were inversely related to $F^-$, $Cl^-$, $SO_4^{-2}$, $Na^+$, and $NH_4^+$, but directly related to turbidity, and total phosphorus. Microcystin-RR concentrations in water samples were inversely related to $Cl^-$, $Br^-$, $NO_3^-$, $SO_4^{-2}$, $Na^+$, $NH_4^+$, conductivity, alkalinity, and TDS, but directly related to $Ca^{+2}$, water temperature, TSS, and turbidity. Cylindrospermopsin and saxitoxin concentrations in water samples were inversely correlated with $Mg^{+2}$ and directly correlated with water temperature.

Overall, water turbidity appeared to be the best predictor of cyanobacterial toxin concentrations; cylindrospermopsin concentrations were lower with increases in turbidity and microcystins and anatoxin were higher with increases in turbidity. The direct relationship between toxin concentrations and turbidity is surprising given that increases in turbidity would reduce light penetration and presumably cyanobacterial growth (*Ota et al., 2015*; *Wells et al., 2015*; *Ashraful Islam & Beardall, 2017*). Perhaps toxin-producing cyanobacteria are out-competing other species under the reduced light conditions. Increases in anatoxin, M-RR, cylindrospermopsin, and saxitoxin concentrations also correlated with increases in water temperature, consistent with higher cyanobacterial growth rates at higher water temperatures (*Butterwick, Heaney & Talling, 2005*; *Watkinson, O'Neil & Dennison, 2005*).

**Table 2  Correlation ($p$ values) between water parameters and cyanobacterial toxin concentrations in 89 water samples collected from a reservoir in the southwest U.S. from September 2015 to September 2017.**

| Parameter | Anatoxin-A | M-LR | M-LA | M-RR | Cylindrospermopsin | Saxitoxin |
|---|---|---|---|---|---|---|
| $F^-$ | ↓ ($9.3E^{-05}$) | NS | ↓ (0.05) | NS | NS | NS |
| $Cl^-$ | ↓ ($1.8E^{-05}$) | NS | ↓ (0.03) | ↓ (0.03) | NS | NS |
| $Br^-$ | ↓ ($8.7E^{-05}$) | NS | NS | ↓ ($5.0E^{-05}$) | NS | NS |
| $NO_3^-$ | ↓ ($2.4E^{-05}$) | NS | NS | ↓ (0.01) | NS | NS |
| $SO_4^{2-}$ | ↓ ($5.6E^{-07}$) | NS | ↓ (0.02) | ↓ (0.01) | NS | NS |
| $Na^+$ | ↓ ($1.4E^{-06}$) | NS | ↓ (0.02) | ↓ (0.003) | NS | NS |
| $NH_4^+$ | ↓ ($5.8E^{-09}$) | NS | ↓ (0.04) | ↓ (0.01) | NS | NS |
| $K^+$ | NS | NS | NS | NS | NS | NS |
| $Mg^{2+}$ | NS | NS | NS | NS | ↓ (0.03) | ↓ (0.01) |
| $Ca^{2+}$ | ↑ ($4.9E^{-08}$) | NS | NS | ↑ (0.02) | NS | NS |
| pH | ↑ (0.01) | NS | NS | NS | NS | NS |
| Temperature | ↑ ($2.6E^{-05}$) | NS | NS | ↑ (0.03) | ↑ (0.01) | ↑ (0.004) |
| Conductivity | ↓ ($2.4E^{-08}$) | NS | NS | ↓ (0.03) | NS | NS |
| Alkalinity | NS | ↓ (0.05) | NS | ↓ (0.03) | NS | NS |
| TDS | ↓ ($3.3E^{-10}$) | NS | NS | ↓ (0.01) | NS | NS |
| TSS | NS | ↑ ($5.0E^{-05}$) | NS | ↑ (0.02) | NS | NS |
| Turbidity | ↑ (0.002) | ↑ ($5.7E^{-05}$) | ↑ (0.01) | ↑ (0.01) | ↓ (0.02) | NS |
| True color | NS | NS | ↓ (0.01) | ↑ (0.03) | ↑ (0.01) | NS |
| DO | ↓ (0.02) | NS | NS | NS | NS | NS |
| Total P | ↑ (0.03) | ↑ ($2.3E^{-05}$) | ↑ (0.01) | NS | NS | NS |
| Location | NS | NS | NS | NS | NS | NS |
| Season | NS | NS | NS | NS | NS | NS |

**Notes.**

NS, not significant.

Significant correlation if $p \leq 0.05$.

↑ indicates direct (positive) correlation.

↓ indicates indirect (negative) correlation.

Nutrients appear to play an important role in cyanobacterial blooms, particularly N and P (*Filstrup & Downing, 2017*). Unfortunately, the relationship between blooms and the presence of cyanobacterial toxins in the water is less consistent. Complicating these relationships (and attempts at correlations) are observations indicating that factors such as salinity not only influence the uptake of P by cyanobacteria (*Markou, Vandamme & Muylaert, 2014*), salinity may also stimulate the release of extracellular toxins (*Preece et al., 2017*).

Water levels, which are affected by seasonal weather changes, also influence cyanobacterial growth and the potential for toxin formation in freshwater ecosystems. In our study, water levels in the lake were 90–100% of capacity; water inputs favoring algal growth were continuous, although inorganic ions did not significantly vary among season and sampling locations. In freshwater lakes, microcystin peaks and toxin transport frequently occur in receiving waters during high flow periods (*Gobler et al., 2016*). Uneven rainfall patterns due to climate change (prolonged droughts followed by intense rainfall) can lead to increased freshwater inputs.

## CONCLUSIONS

Cyanobacterial toxin concentrations during the study period never exceeded safety limits. In addition, toxin concentrations at levels above the limit of quantitation were infrequent. Cylindrospermopsin and saxitoxin were the most frequently detected analytes above the detection limit; their concentrations were directly related to water temperature in the lake and inversely related to $Mg^{+2}$ concentrations in lake water. Microcystin and anatoxin concentrations were inversely correlated with $Cl^-$, $SO_4^{-2}$, $Na^+$, and $NH_4^+$, and directly correlated with turbidity and total P.

## ACKNOWLEDGEMENTS

The authors are grateful to the local health department for the collection of water samples in this study.

### Funding

This work was supported by the Grayson County (Texas) Health Department. The funders had no role in study design, data collection and analysis, decision to publish, or preparation of the manuscript.

### Grant Disclosures

The following grant information was disclosed by the authors:
Grayson County (Texas) Health Department.

### Competing Interests

Todd Anderson is an Academic Editor for PeerJ.

### Author Contributions

- Seenivasan Subbiah, Adcharee Karnjanapiboonwong and Degeng Wang performed the experiments, analyzed the data, prepared figures and/or tables, authored or reviewed drafts of the paper, approved the final draft.
- Jonathan D. Maul conceived and designed the experiments, performed the experiments, contributed reagents/materials/analysis tools, authored or reviewed drafts of the paper, approved the final draft.
- Todd A. Anderson conceived and designed the experiments, performed the experiments, contributed reagents/materials/analysis tools, prepared figures and/or tables, authored or reviewed drafts of the paper, approved the final draft.

### Data Availability

  The raw data are available in the Supplemental File.

## Supplemental Information

Supplemental information for this article can be found online at http://dx.doi.org/10.7717/peerj.7305#supplemental-information.

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
