# Peer review of "Monitoring cyanobacterial toxins in a large reservoir: relationships with water quality parameters"

_PeerJ, doi:10.7717/peerj.7305_

## Round 0.1 · original submission · Major Revisions

Please address all the comments and resubmit it.

Reviewer 1 ·

Basic reporting

This manuscript correlates a good amount of water quality data (cations and anions) with several cyanotoxins in a reservoir in Texas. The data during a two-year monitoring period was extensive to make a strong case and is publishable. However, it appears that the manuscripts did not go through the polishing and scrutiny by the senior author of this manuscript. There are some irrelevant and distracted discussions needed improvement of graphing and language. Since the data is in good quality, I think it can be publishable after a major revision of the writing part.

Experimental design

The work is within the Aims and Scope, it reflects one of the challenges in the harmful algal bloom study. In the Introduction, there should have some more literature review decsribing why the algal toxins could be related to the inorganic ions, and are there any similar work/data on this subject, rather than a general discussion of water quality and cyanotoxins unless the main them of this manuscript is changed to the general water quality and algal toxin issue. I believe the data collection parts and the analytical methods are rigorous with essential details.

Validity of the findings

The authors should present some representative figures for the correlation data instead of the tables with only the p-values such as Table 2 and Table 3. The readers will have a better idea of the correlations between toxins and the anions/cations, which is the main novelty and/or knowledge/data gap of this study.

Conclusions/discussions should also be focused on this main theme rather than the distraction of some general water quality discussions.

When presenting the cationic and anionic data from water quality analysis, it is always prudent to double check the validity of the analysis by checking the charge balance between cations and anions by converting mg/L into meq/L (e.g., data in Table 1). if the charge balance is not closed, then it indicates the analysis of certain ions or the errors in analysis.

The section starts Line 198 should include some figures showing the relationships between water quality and toxins. The scatter plots with indicated p-value are more powerful than just the data shown in Table 2 and Table 3.

Table 1 can be better shown with some box plots?

Table 2: The scientific notation is written incorrectly, should be xxE+05. For K+, it is 87? or 0.87?

In the results section explaining the ones with p<0.05 (Table 3), there should have some more elaboration why Cl-, Na+, Ca2+ have a significant positive correlation with space (location), and why NO3-, Na+, and Mg2+ have a seasonal correlation?

Additional comments

1. From the abstract, it is hard to tell rationales what leads to the main research question regarding the relationship between toxins and common ions. There seems also research question regarding cell count numbers and toxins that confuses the readers. The results part of the abstract is also weak because there is no clear conclusion of such relationship and an indication of what is the implication of such a relationship. Rewrite the abstract.
2. Some typos and grammar errors: (1) lIne 26, parenthesis rather than in its superscript. (2) Line 47-48: hard to understand? (3) Line 44: Add "the" prior to United States, (4) LIne 67-69: Should just be N in nay form dissolved in water rather than N2 (ga) which is the atmosphere, (5) Line 110: the (we" and "they (samples) are awkward; (6) Line 117: why "deep" is relevant here for samples stored in a constant temperature freezer; The same is true on Line 162 (deep-freezer); (6) Line 127: water samples rather than water experimental samples; (7) Line 191: remove the dash before the word "detected"; (8) Line 251: add "were" between "month" and "relatively"; (9) "can start" not "can started"' (9) Line 259: which are subsequently enhanecs? (10) Line 27: "were either retard or stimulate": remove "were"; (11) Line 282: add "that" before patasssium (K); (12 "could be support the" to "could support the" (13) Line 289: add "the" prior to Us EPA. Grammar and language errors like these can be easily eliminated if teh senior authors spent time polishing it.
3. A map showing the area of Grayson and various sampling locations in the Supporting Info should be included, in particular, the results section describe the spatial variation. With such information, the results on spatial variation will be hard for the readers to follow.
4. Line 139-183: Spike recovery studies should be placed in the materials and Methods section,
5. The Discussions starting line 219 is pretty weak, most general water quality discussions are ok, but they have deviated from the major theme of this work, i.e., the relationship between toxins and w\ions. Some statements are a common knowledge, and should be removed, such as Line 252-253; line 269
6. The Conclusion part (Line 288-294) has no conclusion regarding the main topic of this study.

Reviewer 2 ·

Basic reporting

1. The English language and the flow of text should be improved to ensure that an international audience can clearly understand your text.
a) Lines 42-44 " According to Paerl (2008), increasing rate of cyanobacterial hazard algal blooms (cHABs) in aquatic ecosystem have undoubtedly contributed by the loading of nitrogen (N2) and phosphorous (P)."
Here the authors probably mean that the increasing loading of N2 and P have contributed to an increasing rate of cHABs. This sentence needs to be re-written.
b) Lines 49-52 give an introduction to cHABs and should precede Lines 42 so that a reader already has an understanding of the subject before the factors affecting it are described
c) Lines 44-45 describe economic loss due to cHABs. A line describing how they are affecting the economy can help the reader understand the statement.
d) Lines 61 to 68 is a single sentence which describes various factors affecting cyanobacterial growth. This sentence needs to be grammatically corrected or perhaps better if broken into shorter sentences.
e) Line 87 is missing a word " Methanol, acetonitrile, formic acid and water were with LC/MS Optima grade and obtained from Fisher Chemicals (Fair Lawn, NJ, USA)". This sentence should be Methanol, acetonitrile, formic acid and water were analyzed with LC/MS Optima grade and obtained from Fisher Chemicals (Fair Lawn, NJ, USA)
f) Line 125 " Anatoxin-a, microcystin-LR,-LA, and -RR by LC-MS/MS". This should again be something like Anatoxin-a, microcystin-LR,-LA, and -RR analysis by LC-MS/MS
g) Lines 180-184 : These lines also needs grammatical correction. " The recovery of LCS and MSS in algal toxin analysis were fall in the range of 50 to 150 % (US EPA Method 544, 2015) during the study period. Reported results were uncorrected based on recovery. Whereas the observed recovery of anions and cations were fall in the range of 85 to 115 % (US EPA Method 300.1, 1997)."
h) Lines 190 seems to have a typo in reporting BDL of MC and needs to be 0.024 ng/mL instead of 0.0.24 ng/mL
Overall there are numerous grammatical errors throughout the text and it is beyond my scope to point each and every one but I would recommend to do a thorough proof reading and get these corrected during revision.

Experimental design

Lines 122 mention that ELISA tests were performed using method provided by Abraxis Inc. A short summary of the analysis method or at least a reference should be provided.

Validity of the findings

1. In the discussion section "Inorganic ions load, salinity, lake water level and in fresh water ecosystem" the authors state that they " quantified traces (~ 1 mg/L) of NO3- throughout the sampling period". I am not exactly sure what the authors are referring to here. Their collected data doesn't support this statement. Are they referring to average concentration levels throughout the year? Please clarify.
2. Furthermore, in the same discussion section the authors discuss salinity as a variable. I do not see any data related to salinity unless the authors are correlating it to conductance. Please clarify.
3. In the "Algaltoxins and water temperature" section the authors are not describing their results clearly. They state " In our study, traces amounts of microcystins presented during summer seasons and the water temperature values from the EPA database for the corresponding month relatively higher than 15 ˚C. It was supported by the following discussion.". It is unclear what their results are from the above statement. Clearly from their Table 2 they found a positive correlation. The authors should describe this more clearly.
4. In the " Turbidity, pH and conductivity" section the authors state " The turbidity in water system were either retard or stimulate the algal growth. It depends on the availability of nutrients in the water system (Wang, 1974). In clear water systems, light can penetrate and favor for algal growth". Are the authors stating that the turbidity depends on availability of nutrients? Please clarify.

Additional comments

“Monitoring algal toxins in a large reservoir: Relationships with inorganic ions”

The authors monitored cyanobacterial toxin concentrations and inorganic ions in monthly water samples from a large reservoir over a 2+ year period. Seasonal changes in cyanobacterial toxin concentration were observed.

Overall the authors have collected samples and analyzed them adding useful information to the database of the scientific community. The English language can be improved throughout the text for better understanding to the readers. The conclusions can be more specific even if no correlation or inconclusive results are obtained. I would recommend to accept this publication after necessary revisions by the authors.

·

Basic reporting

The manuscript entitled “Monitoring algal toxins in a large reservoir: Relationships with inorganic ions” has described the monitoring cyanobacterial toxin concentrations and inorganic ions in monthly water samples from a large reservoir over a 2+ year period, to regulate the existence of any relationships between toxins and common ions. Cyanobacterial toxins have been quantified using ELISA (cylindrospermopsin, saxitoxin) or LCMS/MS (microcystins, anatoxin-A). Additionally, Common anions and cations have been determined by ion chromatography (IC). Anion (Cl-, NO3-, SO4-2) and cation (Na+, Mg+2, NH4+) concentrations varied by location within the reservoir. According to this reviewer, the present work is good and could be beneficial. This work could be accepted in PeerJ after incorporating the following comments/suggestions.
They are:
1. Please check the manuscript for typographical error.
2. Some relevant references are missing. For better understanding the following references should be cited and discussed in the main body of the manuscript (ACS Omega, 2018, 3, 6624-6634; Inland waters 2017, 7, 385-400; Toxins (Basel). 2015, 7, 1048-1064; J. Environ. Sci. Health A13 (l): 493–499).

Experimental design

na

Validity of the findings

na

Additional comments

The manuscript entitled “Monitoring algal toxins in a large reservoir: Relationships with inorganic ions” has described the monitoring cyanobacterial toxin concentrations and inorganic ions in monthly water samples from a large reservoir over a 2+ year period, to regulate the existence of any relationships between toxins and common ions. Cyanobacterial toxins have been quantified using ELISA (cylindrospermopsin, saxitoxin) or LCMS/MS (microcystins, anatoxin-A). Additionally, Common anions and cations have been determined by ion chromatography (IC). Anion (Cl-, NO3-, SO4-2) and cation (Na+, Mg+2, NH4+) concentrations varied by location within the reservoir. According to this reviewer, the present work is good and could be beneficial. This work could be accepted in PeerJ after incorporating the following comments/suggestions.
They are:
1. Please check the manuscript for typographical error.
2. Some relevant references are missing. For better understanding the following references should be cited and discussed in the main body of the manuscript (ACS Omega, 2018, 3, 6624-6634; Inland waters 2017, 7, 385-400; Toxins (Basel). 2015, 7, 1048-1064; J. Environ. Sci. Health A13 (l): 493–499).

---

## Round 0.2 · accepted · Accept

The manuscript - Monitoring cyanobacterial toxins in a large reservoir: Relationships with water quality parameters - has been Accepted for publication. Congratulations!